# Preventing and Overcoming Resistance to PARP Inhibitors: A Focus on the Clinical Landscape

**DOI:** 10.3390/cancers14010044

**Published:** 2021-12-23

**Authors:** Rosario Prados-Carvajal, Elsa Irving, Natalia Lukashchuk, Josep V. Forment

**Affiliations:** 1DDR Biology, Bioscience, Oncology R&D, AstraZeneca, Cambridge CB4 0WG, UK; mariadelrosario.pradoscarvajal@astrazeneca.com (R.P.-C.); elsa.irving1@astrazeneca.com (E.I.); 2Translational Medicine, Oncology R&D, AstraZeneca, Cambridge CB4 0WG, UK; natalia.lukashchuk@astrazeneca.com

**Keywords:** PARP inhibitor (PARPi), resistance, homologous recombination repair (HRR), molecular mechanisms, clinical relevance

## Abstract

**Simple Summary:**

PARP inhibitors (PARPi) are current treatment options for patients with ovarian, breast, pancreatic or prostate cancer. Although PARPi have transformed the patient journey in these disease settings, resistance eventually develops, leaving them with limited therapeutic opportunities. In this review, we summarize the mechanisms of resistance to PARPi described in pre-clinical models, focusing on the most clinically relevant and proposing ways to tackle them.

**Abstract:**

Poly(ADP-ribose) polymerase (PARP) inhibitors (PARPi) are now a first-line maintenance treatment in ovarian cancer and have been approved in other cancer types, including breast, pancreatic and prostate. Despite their efficacy, and as is the case for other targeted therapies, resistance to PARPi has been reported clinically and is generating a growing patient population of unmet clinical need. Here, we discuss the mechanisms of resistance that have been described in pre-clinical models and focus on those that have been already identified in the clinic, highlighting the key challenges to fully characterise the clinical landscape of PARPi resistance and proposing ways of preventing and overcoming it.

## 1. Introduction

The discovery more than 15 years ago of a synthetic lethal (SL) relationship between mutations in the breast cancer susceptibility, tumour-suppressor genes *BRCA1* and *BRCA2* and the inhibition of poly(ADP-ribose) polymerase (PARP) enzymes [1,2] spearheaded the clinical development of PARP inhibitors (PARPi). Patients with germline mutations in *BRCA1* or *BRCA2* (gBRCAm), when inherited in heterozygosity, have an increased risk of developing cancer, usually of ovarian or breast origin [3]. Especially in the case of gBRCA2m, this also extends to prostate and pancreatic cancer [4,5]. Tumours of these patients almost universally lose both functional copies of the *BRCA1* or *BRCA2* gene, either by loss of heterozygosity (LOH) of the wild-type allele or, more rarely, by the acquisition of a somatic mutation in the wild-type locus [6]. This leads to the loss of function of homologous recombination repair (HRR), the DNA repair pathway in which BRCA proteins play a key role (reviewed in [7]) (Figure 1). An SL relationship between two genes is described when the functional loss of both leads to cell death, while a mutation or defect in either of the two does not greatly impact cell viability. Given that concomitant *BRCA1* and *PARP1* loss showed a clear SL relationship in mouse models [8], it was long assumed that the efficacy of PARPi in BRCAm settings was solely driven by them inactivating PARP enzymatic activity (reviewed in [9]). The discovery that PARPi could also stabilise PARP enzymes on DNA (an effect named “PARP trapping”), and that their efficacy as single agents in BRCAm cell lines was linked not to their potency as enzymatic inhibitors but to their trapping ability [10], represented a turning point in our understanding of the mechanism of action of PARPi. Importantly, all PARPi currently approved as monotherapies are efficacious PARP trappers, although with different levels of potency (reviewed in [11]). Since the approval of the first PARPi, olaparib, in ovarian cancer [12], PARPi have now been approved in all tumour types where gBRCAm are prevalent (Table 1).

Given that BRCA proteins play key roles in HRR, it was proposed very early on in the development of PARPi that they could also be efficacious in non-BRCAm, HRR-deficient (HRD) backgrounds (Figure 1) [13]. Overcoming the limitations of genomic testing, several genetic tests to measure the HRD status of tumours have been developed, with some being approved as companion diagnostic tests for the clinical use of PARPi (Table 1) [14]. HRD testing has identified a sub-population of HRD cancers that benefit from PARPi treatment and cannot be explained by mutations in *BRCA* genes [15,16]. Although defects in other key HRR proteins such as PALB2 account for some of these HRD-positive tumours, it is also important to highlight that some HRR genes, in particular *BRCA1* and *RAD51C*, show high levels of promoter hypermethylation leading to gene silencing in breast and ovarian cancer [17,18]. Thus, it is clear that the use of PARPi can be extended beyond gBRCAm settings.

Late-phase clinical trials with PARPi have shown spectacular, practice-changing results, particularly in ovarian cancer [16,19]. However, as is the case for most targeted therapies, resistance eventually arises, especially when exploring advanced disease settings [20,21]. All mechanisms of resistance described to date can be divided into those where the BRCA functionality (or HRR proficiency) status of the tumour cells plays a key role in determining PARPi response (Figure 2) and those that operate independently of such status. This classification already highlights that only mechanisms of resistance that modulate the BRCA/HRR status of the tumour have been consistently observed clinically or in patient-derived xenograft (PDX) models in relevant disease settings (Figure 3), which will be the main focus of this review.

## 2. BRCA/HRR-Independent Mechanisms of PARPi Resistance

### 2.1. Epithelial–Mesenchymal Transition

Epithelial–mesenchymal transition (EMT) is characterised by the loss of cell–cell interactions and apical–basal polarity and has often been associated with resistance to various therapies [22]. In a study exploring olaparib resistance in a breast BRCA2m genetically engineered mouse model (GEMM), EMT was proposed as the most frequently occurring mechanism of resistance, detected using well validated gene expression changes [23]. In another study using a breast BRCA2m GEMM, a mesenchymal-like tumour subtype was described as less sensitive to PARPi compared to more epithelial tumours [24]. It has also been reported that an EMT gene expression signature is associated with PARPi resistance in small-cell lung cancer (SCLC) PDX models and cell lines. These results must be considered carefully, however, as PARPi are not currently approved in SCLC and the determinants of PARPi responses could vary between different disease settings [25]. Despite these correlative observations, both the mechanism driving EMT-related PARPi resistance and its clinical significance remain unclear.

### 2.2. SLFN11 Loss

Loss of expression of the Schlafen 11 (*SLFN11*) gene is a common feature of human cancer cell lines and provides resistance to DNA-damaging agents, including PARPi [26]. SLFN11 acts in response to DNA damage by halting DNA replication independently of the canonical DNA damage response (DDR) pathway, hence inhibiting cell proliferation [27]. Recent findings linked SLFN11 deficiency with PARPi resistance in SCLC PDX models, although the correlation seemed to depend not only on the SLFN11 status of the models but also on other genetic determinants of response, such as mutations in the DDR kinase ATM [25]. Nevertheless, such a correlation between *SLFN11* expression and PARPi response could not be established in breast cancer PDX models [28]. Interestingly, *SLFN11* downregulation has been recently reported in PARPi progression in two ovarian cancer patients [29]. This highlights the potential differences between disease settings that will be important to be considered when further exploring the relevance of the *SLFN11* gene status as a predictive biomarker of PARPi responses.

### 2.3. P-Glycoprotein Overexpression

Overexpression of the mouse ATP-binding cassette (ABC) drug efflux transporter P-glycoprotein ABCB1, also known as MDR1 in humans, was one of the earlier mechanisms of PARPi resistance to be described in a BRCA1m breast cancer GEMM [30]. Many drugs, including some PARPi, are ABC drug efflux substrates. Upregulation of *MDR1* has been found in small numbers of chemotherapy-resistant and/or PARPi-resistant high-grade serous ovarian cancer patient tumours [29,31]. In the case of PARPi-resistant tumours, *MDR1* overexpression was accompanied by other alterations linked to resistance, suggesting a more complex scenario [29]. These results open the possibility of MDR1 upregulation being a clinically relevant resistance event, particularly in patients heavily pre-treated with chemotherapy before receiving PARPi.

### 2.4. PARG Loss

While PARP proteins catalyse poly(ADP)-ribosylation (PARylation) of their target proteins, a counteracting enzymatic activity is carried out by the poly(ADP-ribose) glycohydrolase, PARG [32]. PARG loss has been associated with PARPi resistance in mouse mammary tumour cell lines and GEMMs of BRCA deficiency. Mechanistically, it has been shown that loss of PARG expression allows for some PARylation to occur even in the presence of PARPi. This includes PARP1 auto-PARylation, which is an important event to allow PARP1 release from DNA. Consequently, PARG deficiency led to reduced PARP1 trapping and DNA damage accumulation [33]. It will be important to determine whether this mechanism of resistance also operates in human tumour samples.

### 2.5. PARP1 Mutations

A clear indication that the primary mechanism of action driving the efficacy of PARPi is their ability to trap PARP1 on DNA came by the identification of PARP1 loss as the strongest determinant of PARPi resistance in mouse embryonic stem cells [34]. This was further confirmed by more recent in vitro studies showing that PARP1 mutant proteins that have lost their ability to bind DNA also confer resistance to PARPi [35,36]. In order to understand the potential clinical relevance of this resistance mechanism, it will be important to determine how compatible this type of PARP1 mutation is with BRCA deficiency given the SL relationship between these genes. Interestingly, a PARP1 mutation predicted to cause loss of DNA-binding ability has been identified in an archival ovarian tumour sample from a patient that did not respond to olaparib. However, it is important to mention that the tumour was platinum resistant and did not harbour gBRCAm (or mutations in other DDR genes) [35], which could be the main reason driving olaparib resistance in this case, regardless of PARP1 status.

## 3. BRCA/HRR-Dependent Mechanisms of PARPi Resistance

### 3.1. Dynamic Biomarkers of HRD

Although genetic testing to characterise gBRCAm and the genomic detection of HRD have proven very useful methods to identify patients that could benefit from PARPi, it is important to highlight that both methodologies rely on the detection of genetic mutations or the assessment of genomic instability (also referred as genomic scars) that are a historical account of the BRCA/HRD status of the tumour. That is, they identify that a tumour was HRD at some point in its evolution, but they do not provide a dynamic measure of such status at the time of treatment. This static nature of genetic and genomic testing could be less relevant in early disease settings or in ovarian cancer, where repeated response to platinum treatment can be used as surrogate biomarker for HRD, but could be problematic when applying these methodologies in late-stage disease [14].

In order to overcome these limitations, there are current efforts to develop functional dynamic biomarkers of HRD. Quantification in formalin-fixed, paraffin-embedded (FFPE) tumour samples of the accumulation of the RAD51 protein in discrete, sub-nuclear structures termed “foci” by immunofluorescence (IF) techniques is one of the most advanced methods [37]. RAD51 is a key mediator of HRR and its recruitment to DNA damage sites by the complex formed by BRCA1–PALB2–BRCA2 proteins is essential for successful DNA repair (Figure 1). This recruitment can be measured by IF using RAD51-specific antibodies and works as a biomarker of HRR proficiency (reviewed in [38]). It has been shown that this method can be used to measure HRD on FFPE sections at baseline without the need to apply exogenous DNA damage to the samples [37], which greatly simplifies its application in clinical material. Interestingly, use of this RAD51 foci assay has highlighted that the great majority of responses to PARPi in PDX and clinical samples analysed to date can be explained by the dynamic HRR status of the tumour in a more accurate way than gBRCAm status or HRD scores, with cases with low RAD51 foci counts predicting a better response to treatment than those with high RAD51 foci counts [37,39,40]. Although most of these analyses were carried out in tumours of breast cancer origin, emerging data suggest that the same could be applied to tumours of prostate [41] or ovarian origin [42]. Whether restoration of RAD51 foci formation is a key mechanism of acquired PARPi resistance in the clinic still remains to be addressed and is challenging due to limited post-PARPi tumour biopsies. However, there have been reports of the restoration of RAD51 foci in breast cancer tumours collected on PARPi progression (in 4/4 patients on PARPi progression, 6/7 patients on PARPi or platinum progression) [43]. In this section describing the mechanisms of PARPi resistance linked to the BRCAm/HRRm status of the tumour, we will highlight whether the discussed mechanism has the potential to restore RAD51 foci formation.

### 3.2. Reversion Mutations

Different to most currently approved precision medicine cancer therapies, PARPi do not target an oncogenic driver event but rather the loss of function of a tumour-suppressor gene. It is thus conceivable to propose that the restoration of the function of the tumour suppressor could provide resistance to PARPi. This was described very early on in the development of PARPi, where secondary mutations in the *BRCA* genes acquired after treatment with platinum or PARPi were identified in vitro and in ovarian cancer patients [44,45,46]. These secondary mutations had the potential to restore the gene’s open reading frame and as such they are referred to as reversion mutations, or simply reversions. Since those first descriptions, *BRCA* reversions have been identified in many tumours from patients progressing on PARPi treatment in all disease settings where PARPi are approved, making them the only clinically validated mechanism of resistance to PARPi described so far [47,48]. Given that ovarian cancer is where PARPi have been approved for longer, it is not surprising that the majority of reversions have been identified in that disease setting, where they account for approximately 25% of the cases of progression after platinum or PARPi treatment [48]. The nature of gBRCAm, which are mostly missense or nonsense mutations or small insertions–deletions leading to frameshifts and premature STOP codons, may explain the prevalence of reversion mutations as a mechanism of resistance. All of them either delete or revert the original mutation and would be predicted to either partially or fully restore BRCA function. Accordingly, reversion mutations have been shown to restore RAD51 foci formation [43]. Importantly, it was recently described that patients harbouring BRCAm involving structural variants such as homozygous deletion of the entire locus, which are inherently resistant to reversion events, are enriched in long-term response groups to PARPi [49].

Reversions have not only been detected in tumours with BRCAm but also in tumours with mutations in other HRR genes such as *PALB2* [50], *RAD51C* and *RAD51D* [51] (Figure 1), which reinforces the notion that mutations outside of *BRCA* genes that impact HRR are valid selection biomarkers for PARPi therapy. As more patients are treated with PARPi outside of ovarian cancer settings, it will be critical to confirm the frequency of reversion mutations driving resistance also in breast, pancreatic and prostate cancer. It is important to emphasize that reversion mutations are often found at low allelic frequencies in all disease settings, highlighting that the improvement of DNA sequencing quality and depth in non-invasive methods to follow cancer progression, such as liquid biopsies, will be key to understanding the true prevalence of BRCA reversions [52].

### 3.3. Restored BRCA/HRR Gene Expression

As *BRCA1* or *RAD51C* gene silencing through promoter methylation has been detected in ovarian and breast tumours [53] and has been associated with HRD cases [54], a potential mechanism of resistance to PARPi in these tumours would involve gene re-expression. Accordingly, analyses of paired biopsies pre- and post-platinum progression of ovarian tumours have shown that the de-silencing of *BRCA1* is linked to platinum resistance [31]. No such correlation has yet been established in post-PARPi clinical progressions, but it has been identified in several cases of acquired PARPi resistance in PDX models of breast [39,55] and ovarian [56] origin. As expected, restored *BRCA1* expression has been correlated with the re-gained ability of the tumour to form RAD51 foci [39]. Interestingly, and in addition to the more trivial mechanistic explanation of de-methylation of the promoter as the cause of re-expression of these genes, de novo gene fusions that place *BRCA1* under the transcriptional control of a heterologous promoter, resulting in its re-expression and the acquisition of therapy resistance, have also been reported [55].

### 3.4. BRCA Hypomorphic Proteins

A hypomorph is a gene or protein variant with similar but weaker effect than the corresponding wild-type version. As mentioned above, reversion mutations in *BRCA* genes leave a genetic trace on the genome that can be used to identify such cases of therapy resistance. As, by definition, reversions are identified in tumours progressing on treatment, it is generally assumed that revertants do not behave as hypomorphic versions of BRCA proteins, at least in fulfilling their function in providing therapy resistance. However, other, mostly non-genetic mechanisms driving BRCA protein re-expression have been described, particularly in the case of BRCA1, and they do involve the generation of hypomorphic variants where entire protein domains can be missing (Figure 4) [57,58,59,60,61]. Although initially these hypomorphs were only described in vitro, there are now several reports of their identification in PARPi-resistant PDX models, where they are a prevalent event linked to the restoration of RAD51 foci formation [37,39]. There are fewer reports on the existence of BRCA2 hypomorphs, and studies linking them to resistance are scarce and only in in vitro settings [62]. It will be important to develop relevant pre-clinical models to test the levels of PARPi resistance that can be achieved in vivo by expressing these hypomorphs, and to develop the ability to detect the presence of BRCA-hypomorphic proteins in tumour samples to assess the prevalence of this resistance mechanism in the clinic.

### 3.5. DDR Rewiring

HRR proficiency as measured by high basal levels of RAD51 foci could be explained by the acquisition of BRCA reversions, re-expression of HRR genes or BRCA-hypomorph activity, all of which have been identified and account for the majority of cases in PDX cohorts [37,39]. However, alternative ways of re-gaining HRR proficiency specifically in BRCA1m cancer cells have been described that do not affect the BRCA1m status of the cell. The best studied mechanism involves the inactivation of the *TP53BP1* gene encoding the 53BP1 protein. The genetic interaction between *BRCA1* and *TP53BP1* operates beyond PARPi resistance settings, and was initially described following the observation that 53BP1 loss rescues the early embryonic lethality caused by BRCA1m in mice [63]. This is due to the suppression of genomic instability caused by BRCA1 loss in the absence of 53BP1, a scenario that extends to the genomic instability caused by PARPi in BRCA1m cells [64,65]. Subsequent work by many different research groups has highlighted that 53BP1 does not operate in isolation with regard to generating resistance to PARPi in BRCA1m settings, and seems to act as the central component of a protein complex known as 53BP1–Shieldin (Figure 2; reviewed in [66]). Loss of 53BP1–Shieldin complex components results in the restoration of RAD51 foci formation and a re-gained ability to perform HRR in the absence of BRCA1. This seems to be mostly due to 53BP1–Shieldin blocking the alternative, BRCA1-independent pathway of recruitment of PALB2–BRCA2–RAD51, but also by preventing the nucleolytic processing of DNA ends (a phenomenon called DNA-end resection), which is an essential upstream event in HRR [67,68,69] (Figure 1). Importantly, mutations in 53BP1 have been identified in PDX models on progression on PARPi treatment and linked to restored RAD51 foci formation [37], and also in one paired clinical sample on PARPi progression [43]. As all of these examples have been identified in breast cancer, it will be important to understand their prevalence in other disease settings and the clinical relevance of mutations in components of the 53BP1–Shieldin complex other than 53BP1 itself. Moreover, it will be interesting to explore whether DDR rewiring can also occur in other HRD tumours, as 53BP1–Shieldin mutations specifically confer resistance to PARPi in BRCA1m settings [69].

### 3.6. Restoration of Replication Fork Protection

In addition to their canonical roles in HRR (Figure 1), BRCA proteins are also key players in a form of DNA replication fork protection (RFP) that prevents stalled and regressed replication forks from being degraded by the action of DNA nucleases [70,71]. As such, deficiency in the recruitment of these nucleases to stalled replication forks or the defective remodelling of the forks that is required for their processing have been shown to cause a moderate level of PARPi resistance in BRCAm cell lines (reviewed in [72]). The relevance of the restoration of RFP as a driver of PARPi resistance, however, is disputed by the fact that (i) the separation of function mutations in BRCA1 [73] or BRCA2 [70] that affect their RFP function but leave their HRR role intact do not confer sensitivity to PARPi, and (ii) loss of 53BP1, which causes PARPi resistance in BRCA1m settings, restores RAD51 foci formation and HRR but not RFP [74]. Accordingly, this potential mechanism of resistance has only been described in vitro, and further accumulation of in vivo data will be required to assess its importance in clinically relevant settings.

## 4. Preventing and Tackling PARPi Resistance

Although tumours may eventually develop PARPi resistance, the current data suggest that, as is the case with many other therapies, earlier intervention results in more durable responses regardless of the tissue of origin. The generally positive safety profile of PARPi allows their use in the maintenance setting, where they have provided the most extraordinary results. In ovarian cancer, for example, first-line maintenance treatment trials with olaparib showed an estimated median progression-free survival of more than 3 years, compared to 19.1 months in the second line setting [19,75]. Increased benefit through earlier intervention has also been observed in breast cancer [76,77] and there are ongoing trials to assess this in prostate cancer [21,78].

As PARPi are approved in first-line maintenance settings, efforts should not only be directed towards tackling resistance once it arises but also to prevent it appearing in the first place. Any approach that would enhance the efficacy of PARPi while sparing the healthy tissue will help in this regard. Pre-clinical work suggested that combinations of PARPi with immune-oncology agents should be considered owing to the superior efficacy observed for PARPi in immune-competent mice [79,80,81], something that is currently being tested clinically in many different disease settings (reviewed in [82]). It was also reported that inhibition of the vascular endothelial growth factor (VEGF) causes a certain level of HRD that could be further exploited by PARPi, but results from the Phase III PAOLA-1 trial showed the benefit of the combination of olaparib and the VEGF inhibitor bevacizumab only in the HRD population [15]. Similarly, inhibition of the phosphoinositide 3-kinase (PI3K) pathway has been shown to impact the expression of HRR genes, thus highlighting the potential value of PI3K pathway inhibition combined with PARPi in non-HRD tumours [83,84]. Moreover, a cross-talk between PARP1 and androgen receptor (AR) function has been proposed as the basis of the benefit of the combination of PARPi with AR inhibitors regardless of the HRD status of the tumour [85,86,87]. Clinical trials exploring these combinations in unselected populations are already under way and will shed light into the validity of the approach [78,88,89]. Given the success of antibody–drug conjugates (ADCs; reviewed in [90]) harbouring a DNA topoisomerase I (TOP1) inhibitor warhead in breast cancer [91], and the known synergistic interaction between TOP1 inhibitors and PARPi [92], combinations of TOP1i-ADCs with PARPi should also be considered. More recently, an SL interaction between BRCAm and loss of the key microhomology-mediated end joining (MMEJ) DNA repair factor DNA polymerase theta (also known as POLQ) has been described [93,94]. As the first POLQ inhibitors [95] are entering the clinic, it will be interesting to assess their activity when combined with PARPi.

In all cases, however, patient relapse after PARPi treatment will usually leave them with few established therapeutic options other than platinum re-challenge followed by repeated PARPi maintenance, a regime that has shown limited benefit in ovarian cancer [96]. As described above, the most obvious differentiation between resistance mechanisms is whether tumours restore HRR or remain HRD. Accordingly, the potential therapeutic options that we discuss below can also be linked to the HRD status of the tumour at the time of treatment (Figure 5).

### 4.1. Non-HRD Tumours

This represents the best clinically described group, as it includes tumours with reversion mutations or with loss of HRR gene promoter methylation. These cases will be, in general, also resistant to platinum treatment (Figure 5). As such, therapeutic options will have to rely on attacking alternative vulnerabilities already present in the primary tumour or acquired on PARPi progression. It has been proposed that PARPi-resistant tumours accumulate high levels of replication stress, a cell status defined by sub-optimal conditions for DNA replication (reviewed in [97]). The acquisition of amplifications in the *CCNE1* gene, encoding the cyclin E protein, in ovarian tumours resistant to PARPi point towards that direction [29]. Cells rely on a signal transduction pathway termed the replication stress response (RSR) to deal with high levels of replication stress, and drugs inhibiting its key effector kinases ATR, CHK1 and WEE1 are already in clinical trials (reviewed in [98]). It has been shown in PDX and xenograft models that combinations of PARPi with ATR or WEE1 inhibitors can be effective treatment options in PARPi-resistant models [99,100], something already being tested in clinical trials with encouraging reported efficacy [101,102,103] (and NCT04197713). These combinations could also be a good option to consider even in PARPi-naïve settings, as a way to increase efficacy and reduce the chances of residual disease accumulation, but tolerability margins should be carefully managed [100]. In general, patient selection strategies to identify PARPi-resistant tumours with high levels of replication stress will be key for the success of these approaches.

As discussed before, the synergistic interaction between TOP1i and PARPi provides rationale for the combination of TOP1i-ADCs and PARPi as a way to increase efficacy in earlier lines of treatment, independently of the HRD status of the tumour. This should also be considered in post-PARPi settings, and will be of particular importance in scenarios where tumours have regained HRR proficiency, as is the case here. Key to the success of this approach will be the identification of relevant antigens to target in the post-PARPi populations.

To increase the durability of the treatment, combinations with therapies that could prevent the acquisition of these resistance mechanisms are also attractive. Sequence analyses of reversion events identified in the clinic have highlighted the role that error-prone DNA repair pathways, such as non-homologous end joining (NHEJ) and MMEJ, play in generating them [47,48]. Given that NHEJ and MMEJ inhibitors are entering the clinic [95,104,105], their potential combination with PARPi to prevent the acquisition of reversion mutations should be considered. In addition, there is little mechanistic understanding of the pathways driving the loss of *BRCA1* or *RAD51C* promoter methylation. Identifying the responsible de-methylase(s) will unveil interesting new target opportunities.

### 4.2. BRCAm, HRR-Proficient Tumours

This group mainly represents resistance driven by BRCA-hypomorph expression or DDR rewiring events (Figure 5). The clinical development of a dynamic biomarker of HRD, such as RAD51 foci quantification, will greatly increase our understanding of the level of HRD remaining in tumours re-gaining HRR without losing their BRCAm status. Pre-clinical work has shown that mutations in 53BP1–Shieldin, while causing PARPi resistance, do not result in acquired resistance to platinum treatment [106,107] (although other reports suggest otherwise [65]), highlighting an already available treatment option. Mutations in 53BP1–Shieldin have also been shown to result in an SL interaction with the key MMEJ factor POLQ, offering an acquired therapeutic vulnerability that could be exploited [95]. Furthermore, it has been shown that the restored HRR functionality caused by 53BP1–Shieldin loss in BRCA1m settings relies on sustained signalling through the ATR kinase [68,108], providing support for the idea that combining PARPi and ATR inhibitors could be beneficial in a range of scenarios.

Although BRCA1 hypomorphs have been shown to provide resistance to PARPi in PDX models, it will be important to determine their prevalence in clinical samples, an effort that will probably require the clinical development of non-genetic detection methods [37]. In addition, BRCA hypomorphs have been shown to provide resistance to PARPi in vitro, but it is not entirely clear if the extent of PARPi resistance they provide matches that caused by reversion mutations, and whether they also fully restore platinum resistance. It is also important to highlight that BRCA1 hypomorphs lack entire protein domains that could be important for other BRCA1 functions, such as RFP, which could potentially be exploited with RSR inhibitors. Further pre-clinical investigation of the functions and vulnerabilities of BRCA hypomorphs is required to advance our understanding of how to tackle this resistance mechanism.

### 4.3. HRD Tumours

Although this group encompasses mechanisms that in general will also result in platinum resistance, the fact that the tumours would remain HRD opens the possibility of exploiting other SL interactions that have been described for BRCAm/HRD, with POLQ inhibitors being the most clinically advanced option (Figure 5) (reviewed in [109]). Platinum re-challenge would remain an option in the case of mutations affecting PARP metabolism and, in the specific case of *SLFN11* deficiency, the opportunity of combining RSR inhibitors with standard chemotherapies has already shown promising results [28]. Further pre-clinical work is required to better understand the SL landscape in BRCAm cell lines that have restored their RFP capability without impacting HRR.

## 5. Conclusions

Although a wide range of PARPi resistance mechanisms has been described in pre-clinical models, actual clinical data are scarce and mostly confirm the prevalence of reversion mutations as a primary driver of PARPi failure. This lack of clinical data highlights the need to evaluate PARPi resistance in post-PARPi tumour biopsies. One way to do this would be by increasing the number of clinical trials in the post-PARPi patient population with mandatory biopsies on enrolment, as it will be key to have a dynamic measure of the tumour HRD status at the time of treatment to provide the best therapeutic options going forward. In addition, this will also reveal the true diversity of resistance mechanisms in patients. Access to such samples, together with improvements in the sensitivity of new technologies, such as DNA sequencing in both solid and liquid biopsies, and non-genetic methods of detection of resistance, such as protein biomarkers or promoter methylation status, will enable the focusing of pre-clinical and drug development efforts on the most relevant PARPi resistance mechanisms.

## Figures and Tables

**Figure 1 cancers-14-00044-f001:**
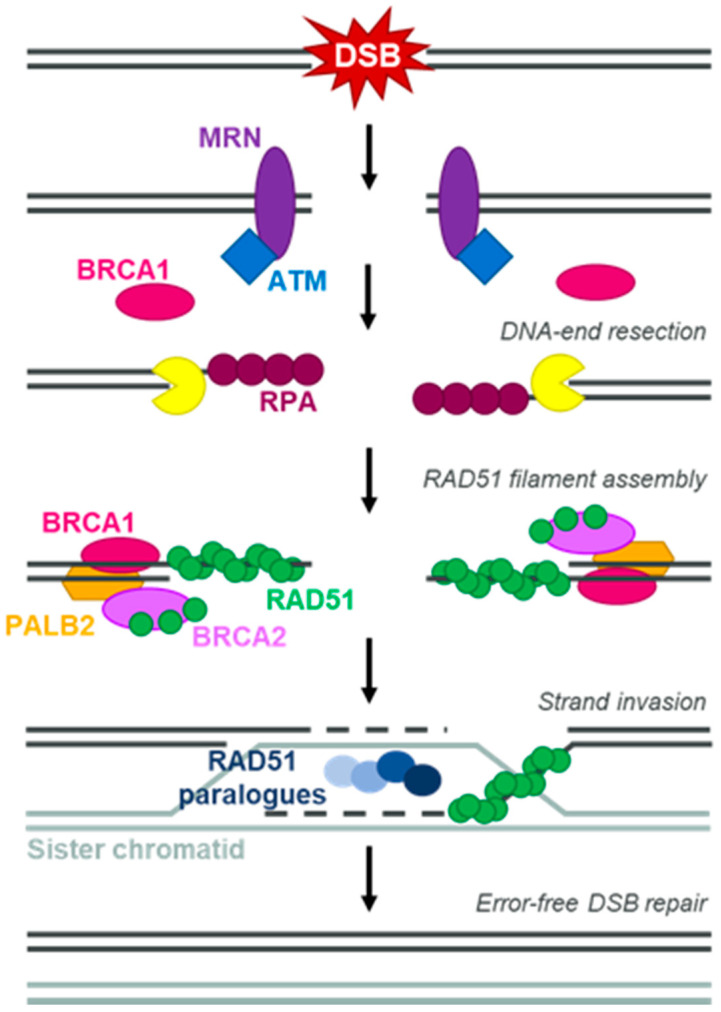
Homologous recombination repair. Simplified schematic of homologous recombination repair (HRR) of a DNA double-strand break (DSB) using the sister chromatid as repair template. Key proteins involved are highlighted in the figure.

**Figure 2 cancers-14-00044-f002:**
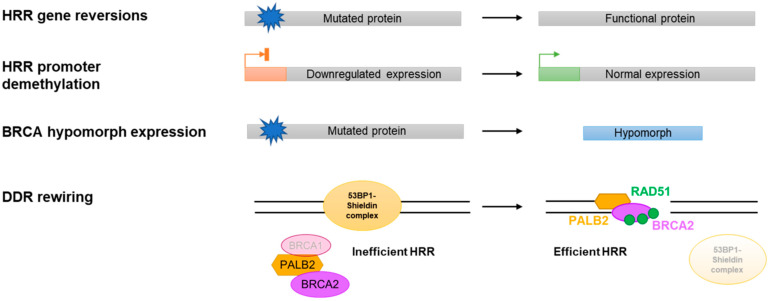
Representation of the mechanisms of PARPi resistance that depend on the HRR status of the cell.

**Figure 3 cancers-14-00044-f003:**
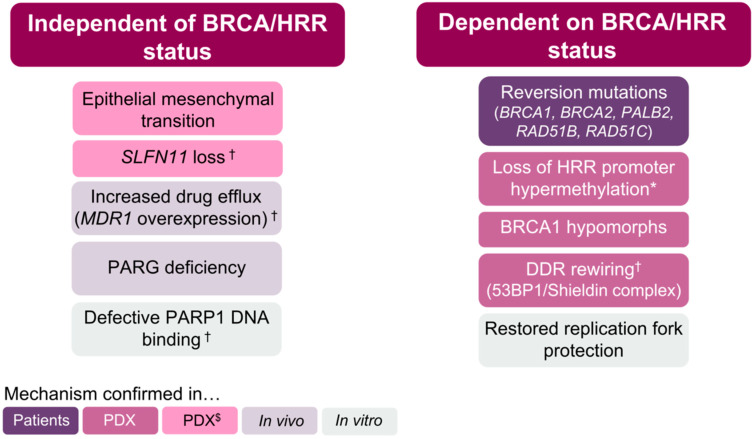
Mechanisms of resistance to PARPi. * also identified in patients with platinum-resistant tumours; ^†^ also reported in one patient; ^$^ identified in models derived from tumour types where PARPi are not currently approved.

**Figure 4 cancers-14-00044-f004:**
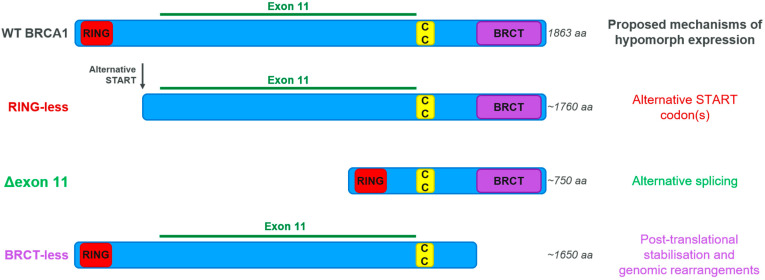
BRCA1 hypomorphs. RING: Really Interesting New Gene domain, with ubiquitin E3 ligase activity. CC: Coiled-Coil region, where BRCA1 interacts with PALB2–BRCA2–RAD51. BRCT: BRCA1 C-Terminus domain, a multi-protein-interacting region.

**Figure 5 cancers-14-00044-f005:**
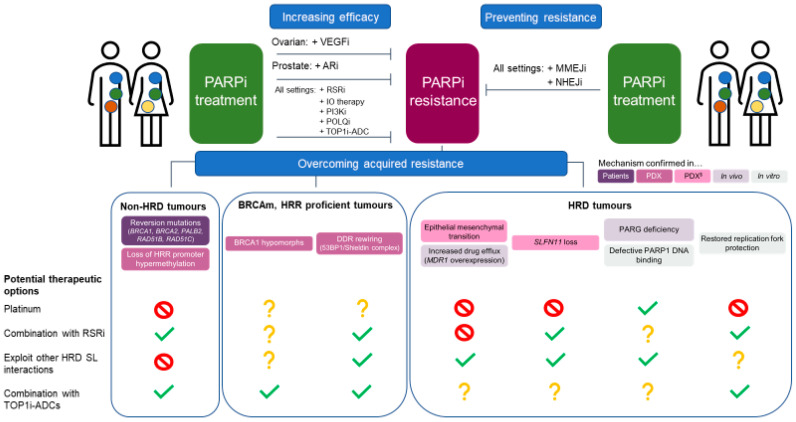
Preventing and tackling PARPi resistance. VEGFi: vascular endothelium growth factor inhibitor; ARi: androgen receptor inhibitor; RSRi: replication stress response inhibitor; IO: immunoncology; ADC: antibody–drug conjugate; PI3Ki: phosphoinositide 3-kinase pathway inhibitor; MMEJi: microhomology-mediated end-joining inhibitor; NHEJi: non-homologous end-joining inhibitor; HRD: homologous recombination deficiency; PDX: patient-derived xenograft; SL: synthetic lethality; HRR: homologous recombination repair.

**Table 1 cancers-14-00044-t001:** US Food and Drug Administration labels for different PARPi.

PARP Inhibitor	Olaparib (Lynparza)–AstraZeneca		Rucaparib (Rubraca)—Clovis Oncology	Niraparib (Zejula)—GSK	Talazoparib (Talzenna)—Pfizer
Cancer type	Monotherapy	Combination	Monotherapy	Monotherapy	Monotherapy
Ovarian	Treatment setting—patients with recurrent gBRCAm advanced cancer who have been treated with 3L+ of chemotherapy Maintenance setting—patients in CR or PR to platinum-based chemotherapy (recurrent disease) and germline or somatic BRCAm advanced cancer (1L)	Maintenance setting—with bevacizumab (VEGFi) in patients in CR or PR to platinum-based chemotherapy and HRD-positive status	Treatment setting—patients with BRCAm (germline and/or somatic) cancer who have been treated with 2L+ of chemotherapies Maintenance setting—patients with recurrent cancer who are in a CR or PR to platinum-based chemotherapy	Treatment setting—patients with advanced cancer who have been treated with 3L+ of chemotherapy and whose cancer is associated with HRD Maintenance setting—patients with advanced cancer who are in a CR or PR to 1L+ platinum-based chemotherapy	N/A
Breast	Treatment setting—patients with gBRCAm, HER2-negative mBC who have been treated with chemotherapy	None	N/A	N/A	Treatment setting—patients with gBRCAm, HER2-negative locally advanced or mBC
Pancreatic	Maintenance setting—patients with gBRCAm mPA whose disease has not progressed on at least 16 weeks of 1L platinum-based chemotherapy	None	N/A	N/A	N/A
Prostate	Treatment setting—patients with germline or somatic HRR gene-mutated mCRPC who have progressed following prior treatment with enzalutamide or abiraterone	None	Treatment setting—patients with BRCAm (germline and/or somatic)-associated mCRPC who have been treated with androgen receptor therapy and a taxane-based chemotherapy	N/A	N/A

Full label documents can be accessed here: Lynparza: https://www.accessdata.fda.gov/drugsatfda_docs/label/2020/208558s014lbl.pdf (accessed on 4 October 2021); Zejula: https://www.accessdata.fda.gov/drugsatfda_docs/label/2020/208447s015s017lbledt.pdf (accessed on 4 October 2021); Rubraca: https://www.accessdata.fda.gov/drugsatfda_docs/label/2020/209115s004lbl.pdf (accessed on 4 October 2021); Talzenna: https://www.accessdata.fda.gov/drugsatfda_docs/label/2018/211651s000lbl.pdf (accessed on 4 October 2021). Abbreviations used: gBRCAm: deleterious or potentially deleterious germline mutation in the BRCA1 or BRCA2 gene; 1L, 2L+, 3L+: first-line, second-line or more, and third-line or more of any given treatment; CR: complete response; PR: partial response; mBC: metastatic breast cancer; mPA: metastatic pancreatic adenocarcinoma; HRR: homologous recombination repair; mCRPC: metastatic, castration-resistant prostate cancer; VEGFi: inhibitor of vascular endothelial growth factor; HRD: homologous recombination deficient; N/A: not approved.

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
