# Peer review of "Preventing and Overcoming Resistance to PARP Inhibitors: A Focus on the Clinical Landscape"

_cancers, 2021, doi:10.3390/cancers14010044_

Round 1

Reviewer 1 Report

The review paper submitted to Cancers on prevention of PARPi resistance and strategies to overcome resistance, presents condensed information on mechanisms of cancer resistance to current PARPi. Introduced more than 15 years ago, PARPi target specifically BRCA1/BRCA2 mutant cancer cells, or HRD (homologous recombination deficient) tumors and revolutionized cancer therapy. Overall, the paper presents concise and current information, and adds a molecular perspective to PARPi use in cancer treatment. Its main focus is on tumor inherent and acquired PARPi resistance, determined both in vitro and in vivo, in various tumor types, and reported in patients. The review includes a key summary from several studies, that could be very useful for researchers or medical practitioners. Therefore paper is recommended for publication after minor editorial revisions.

Several mistakes need to be corrected prior publication:

  1. Figure 1 lacks the title.
  2. Authors should revise carefully references listed in text. Starting from at least reference 47, every other reference is mismatched.
  3. The references in paragraph titled 2.1 Epithelial-mesenchymal transition do not support the information on SCLC, small cell lung cancer, treated with PARPi and the statements provided. In addition in this section formatting of the references is different from the rest of the paper.
  4. In the same paragraph 2.1 Epithelial-mesenchymal transition authors repeat twice the following sentences. “It has also been reported that an EMT gene expression signature is associated with PARPi resistance in small cell lung cancer (SCLC) PDX models and cell lines. These results must be considered carefully, however, as PARPi are not currently approved in SCLC and the determinants of PARPi responses could vary between different disease settings4”. This is unnecessary and needs to be corrected.
  5. In the 2.1 paragraph authors do not provide any additional information on breast cancer that could undergo EMT, and thus become resistant to PARPi. It is recommended to provide additional to small cell lung cancer example on developing EMT by tumor, as it is more relevant for PARPi use in clinic, in breast or ovarian cancer patients. Here is an example of report, publication by Ordonez LD et al., 2019, Oncotarget, “Rapid activation of epithelial-mesenchymal transition drives PARP inhibitor resistance in Brca2-mutant mammary tumors.” https://doi.org/10.18632/oncotarget.26830, PMID: 31080552
  6. In section 4.2 BRCAm, HRR proficient tumors, DDR rewiring events are not represented in Figure 4, but examples of hypomorphs of BRCAm (reverse mutations) are introduced in Figure 3. Please clarify what Figure the authors want to recall in order to support the existence of reversed BRCA mutations, called hypomorphic BRCA versions, what adds to mechanism for PARPi resistance.
  7. The authors only briefly include the combination therapies as a strategy for overcoming resistance to PARPi. It is recommended to include additional paragraph on use of standard ionizing radiation in combination with PARPi for cancer therapy. Review on ionizing radiation and PARPi combination could be found in Lesueur P. et al., Oncotarget 2017, “Poly-(ADP-ribose)-polymerase inhibitors as radiosensitizers: a systematic review of pre-clinical and clinical human studies”, https://www.ncbi.nlm.nih.gov/pmc/articles/PMC5620324/

Author Response

The review paper submitted to Cancers on prevention of PARPi resistance and strategies to overcome resistance, presents condensed information on mechanisms of cancer resistance to current PARPi. Introduced more than 15 years ago, PARPi target specifically BRCA1/BRCA2 mutant cancer cells, or HRD (homologous recombination deficient) tumors and revolutionized cancer therapy. Overall, the paper presents concise and current information, and adds a molecular perspective to PARPi use in cancer treatment. Its main focus is on tumor inherent and acquired PARPi resistance, determined both in vitro and in vivo, in various tumor types, and reported in patients. The review includes a key summary from several studies, that could be very useful for researchers or medical practitioners. Therefore paper is recommended for publication after minor editorial revisions.

We thank the reviewer for his/her overall positive assessment of our Review and for his/her recommendation for publication with minor editorial revisions.

Several mistakes need to be corrected prior publication:

  • Figure 1 lacks the title.

We apologise for not having included one, a defect that is corrected in the current version.

  • Authors should revise carefully references listed in text. Starting from at least reference 47, every other reference is mismatched.

We apologise for this formatting mistake, which we have corrected in the new version.

  • The references in paragraph titled 2.1 Epithelial-mesenchymal transition do not support the information on SCLC, small cell lung cancer, treated with PARPi and the statements provided. In addition in this section formatting of the references is different from the rest of the paper.

Please see answer to the previous point.

  • In the same paragraph 2.1 Epithelial-mesenchymal transition authors repeat twice the following sentences. “It has also been reported that an EMT gene expression signature is associated with PARPi resistance in small cell lung cancer (SCLC) PDX models and cell lines. These results must be considered carefully, however, as PARPi are not currently approved in SCLC and the determinants of PARPi responses could vary between different disease settings4”. This is unnecessary and needs to be corrected.

We apologise for this formatting mistake, which we have corrected in the new version.

  • In the 2.1 paragraph authors do not provide any additional information on breast cancer that could undergo EMT, and thus become resistant to PARPi. It is recommended to provide additional to small cell lung cancer example on developing EMT by tumor, as it is more relevant for PARPi use in clinic, in breast or ovarian cancer patients. Here is an example of report, publication by Ordonez LD et al., 2019, Oncotarget, “Rapid activation of epithelial-mesenchymal transition drives PARP inhibitor resistance in Brca2-mutant mammary tumors.” https://doi.org/10.18632/oncotarget.26830, PMID: 31080552

The reference provided by the reviewer was already included in our original submission (number 23 in the reference list) and mentioned in the following sentence: “In a study exploring olaparib resistance in a breast BRCA2m genetically-engineered mouse model (GEMM), EMT was proposed as the most frequently occurring mechanism of resistance, detected using well validated gene expression changes [23]”.

  • In section 4.2 BRCAm, HRR proficient tumors, DDR rewiring events are not represented in Figure 4, but examples of hypomorphs of BRCAm (reverse mutations) are introduced in Figure 3. Please clarify what Figure the authors want to recall in order to support the existence of reversed BRCA mutations, called hypomorphic BRCA versions, what adds to mechanism for PARPi resistance.

DDR rewiring events were included in Figure 4 (third column), alongside BRCA1 hypomorphs. As several BRCA1 hypomorphs have been described in the literature, the purpose of Figure 3 is to provide a quick and visual guide of how these hypomorphs look like.

  • The authors only briefly include the combination therapies as a strategy for overcoming resistance to PARPi. It is recommended to include additional paragraph on use of standard ionizing radiation in combination with PARPi for cancer therapy. Review on ionizing radiation and PARPi combination could be found in Lesueur P. et al., Oncotarget 2017, “Poly-(ADP-ribose)-polymerase inhibitors as radiosensitizers: a systematic review of pre-clinical and clinical human studies”, https://www.ncbi.nlm.nih.gov/pmc/articles/PMC5620324/

We thank the reviewer for pointing this out but we believe this is out of scope for this Review. Our interest for this Review has focused on the use of PARPi as monotherapy or in combination with other targeted therapies (immune-oncology agents, VEGF inhibitors, ADCs…) and have left combinations with chemotherapies or radiotherapies to be discussed in other Reviews, like the one pointed out by the reviewer.

Reviewer 2 Report

In the manuscript entitled "Preventing and overcoming resistance to PARP inhibitors: a focus on the clinical landscape", the authored tried to summarize the current understanding the treatment of cancers with PARP inhibitors. The autors first introduced the mechanisms mediating the resistance of PARPi treatment, including BRCA/HRR-independent and dependent mechanisms, followed by discussion of preventing and tackling PARPi resistance in different cancers. Overall, this manuscript was well written and presented. I have a few minor concern: 

1) what are the side effects of PARPi treatment? 

2) What kinds of patients are suitable for the treatment of PARPi? Do  they need genetic test before treatment?

3) Combination treatment should be a good way to overcome the resistance. Can the author elaborate more for current studies on this topic?

4) It is hard to read for Figure 1b. Please make it more clearer.

Author Response

In the manuscript entitled "Preventing and overcoming resistance to PARP inhibitors: a focus on the clinical landscape", the authored tried to summarize the current understanding the treatment of cancers with PARP inhibitors. The autors first introduced the mechanisms mediating the resistance of PARPi treatment, including BRCA/HRR-independent and dependent mechanisms, followed by discussion of preventing and tackling PARPi resistance in different cancers. Overall, this manuscript was well written and presented. I have a few minor concern:

We thank the reviewer for his/her overall positive assessment of our Review

1) what are the side effects of PARPi treatment?

As this Review focuses on mechanisms of action and resistance, we feel that this is a question best addressed in other, more pharmacologically-oriented reviews. We do, however, reference the currently approved labels for all PARPi in Table 1, where described side effects can be found.

2) What kinds of patients are suitable for the treatment of PARPi? Do they need genetic test before treatment?

We discuss at several points in this Review what the different requirements in terms of genetic testing are for a patient to be deemed suitable for treatment with PARPi. A summary of these requirements in the different tumour types where PARPi are approved can be found in Table 1.

3) Combination treatment should be a good way to overcome the resistance. Can the author elaborate more for current studies on this topic?

We extensively discuss combination treatment options with other targeted therapies in section 4 (Preventing and tackling PARPi resistance). These combinations are also highlighted in Figure 4 and aligned to specific resistance mechanisms. We feel that combinations with chemotherapies or radiotherapy are out of the scope of this Review.

4) It is hard to read for Figure 1b. Please make it more clearer.

We apologise for this and have now split the two panels in Figure 1 into two different figures (1 & 2 in the new version of the manuscript) to make them bigger and easier to read.